# Epigenetics and Modulations of Early Flavor Experiences: Can Metabolomics Contribute to Prevention during Weaning?

**DOI:** 10.3390/nu13103351

**Published:** 2021-09-24

**Authors:** Angelica Dessì, Alice Bosco, Roberta Pintus, Giulia Picari, Silvia Mazza, Vassilios Fanos

**Affiliations:** Department of Surgical Sciences, University of Cagliari and Neonatal Intensive Care Unit, AOU Cagliari, 09124 Cagliari, Italy; angelicadessi@unica.it (A.D.); alicebosco88@gmail.com (A.B.); giulia.picari@gmail.com (G.P.); siella.mazza@gmail.com (S.M.); vafanos@tiscali.it (V.F.)

**Keywords:** epigenetics, metabolomics, flavor, weaning, complementary feeding

## Abstract

The significant increase in chronic non-communicable diseases has changed the global epidemiological landscape. Among these, obesity is the most relevant in the pediatric field. This has pushed the world of research towards a new paradigm: preventive and predictive medicine. Therefore, the window of extreme plasticity that characterizes the first stage of development cannot be underestimated. In this context, nutrition certainly plays a primary role, being one of the most important epigenetic modulators known to date. Weaning, therefore, has a crucial role that must be analyzed far beyond the simple achievement of nutritional needs. Furthermore, the taste experience and the family context are fundamental for future food choices and can no longer be underestimated. The use of metabolomics allows, through the recognition of early disease markers and food-specific metabolites, the planning of an individualized and precise diet. In addition, the possibility of identifying particular groups of subjects at risk and the careful monitoring of adherence to dietary therapy may represent the basis for this change.

## 1. Introduction

In recent decades, the epidemiological landscape has radically changed throughout the world. The significant reduction in acute pathologies was followed by an equally dramatic increase in chronic non-communicable diseases (NCDs). They include various and complex pathological entities, including diabetes, cardiovascular disease, obesity, hypertension, neoplastic and neuro-degenerative diseases. These data are alarming, just to think that, to date, non-communicable diseases represent one of the main causes of death in the world [1] and that, among these, the one that most affects the pediatric population is obesity [2]. In this regard, it is important to consider that there are particular time windows of vulnerability that can affect the future well-being of an organism. In fact, the first epochs of life, when the cells are undergoing differentiation and the various tissues are developing, are characterized by a high plasticity, essential for allowing the organism to adapt to the surrounding environment [3]. This adaptation is possible thanks to particular molecular mechanisms that control the gene expression through the induction of specific phenotypes without modifying the DNA [4]. Thus, there is a static gene pool and a dynamic phenotype in perfect synergy with the surrounding environment. These epigenetic modifications, although reversible, can be transmitted from one cell generation to the next and, if germ cells are involved, a trans-generational shift can occur. In this context, nutrition plays a fundamental role in many ways. First of all, nutrition is an essential source of key nutrients for numerous epigenetic pathways [1]. Furthermore, excessive weight gain in the first thousand days of life [5] and an excessive protein intake [6] are some of the most important risk factors for obesity. In this regard, the peculiar period of complementary feeding becomes a unique opportunity to set up correct eating habits from an early age and to start the taste education, both key factors for future food choices. Given the strong correlation between nutrition and epigenetics, thus, between food and phenotype, there is an absolute need to take into account the extreme inter-individual variability. Metabolomics represent one of the most functional new technologies for this purpose [7]. Indeed, a subject’s metabolite profile constitutes its metabolic phenotype, meaning the metabotype; undoubtedly, it is closely related to genetics, but it is also influenced by diet and eventual pathologies, since it reflects the health of an individual [8]. Therefore, one can consider the metabolomic profile of a subject as a highly predictive phenotype as it is the result of epigenetic modulation. In this context, considering that nutrition is the epigenetic factor *par excellence*, especially in the early stages of development, prevention measures can be expanded not only through more correct food choices but also thanks to increasingly personalized diagnostic investigation techniques for each subject. The aim is to increase the clinical benefits for each patient with early intervention times, exploiting the speed of metabolic variations [9].

## 2. Epigenetics and Weaning Diet

The introduction of complementary feeding is a very delicate moment in which, on the one hand, future food preferences are forged and, on the other, the high nutritional requirements of the child are met. Beluska-Turkan et al. [10], an expert in the nutritional field, in 2019 performed an extensive review of all the literature in order to determine the essential nutrients in the first thousand days of life. This important study highlighted how, from six months, milk alone is no longer able to meet the increased nutritional requirements of children. In fact, some nutrients play a key role such as fats, including essential fatty acids such as alpha-linolenic acid and linoleic acid. The demands for certain micronutrients such as iron and zinc are also crucial: the requirements of which per kilo of body weight are greater in this period than at any other time in life. In the choice of foods to offer to children to meet their high energy needs, another factor to pay attention to, is the high energy density. The use of green leafy vegetables, legumes and dairy products is suggested, thanks to the presence of calcium, vitamin D and proteins, to promote proper bone development and as a prevention for rickets. It also emerged that, although there is no reference daily intake (RDI), for some carotenoids, such as lutein and zeaxanthin, their consumption is important for correct neurological and visual development and to promote good health in general, as well. Other vitamins, such as vitamins A, C, B6, B12 and folates are very important in this time window, both to prevent major nutritional deficiencies and to promote the absorption of non-heme iron, to support healthy growth and a proper development.

It is essential to assess the nutritional impact of weaning in order to meet the energy requirements and reference daily intakes as well as it is very important to study the correlation with the phenotype, especially with a view to preventive and individualized medicine. The description of the epigenetic mechanisms is beyond the scope of this review. In literature, there are excellent papers concerning this topic [11,12,13] and the main epigenetic mechanisms are illustrated in Figure 1.

Research has shown that all of the phenotype changes are even more important during critical developmental periods, such as the first thousand days of life. Indeed, the extreme plasticity due to the rapid growth of the organism, makes the period from conception to two years of age a window of greater vulnerability [14]. In this context, the role of nutrition is extremely important, with short and long-term repercussions [3], as demonstrated nearly a decade ago by the groundbreaking study of Waterland and Jirtle [15].

However, as highlighted by Randunu et al. [1], the studies concerning postnatal dietary interventions and their epigenetic effects in the short and long term are scarce. In fact, although the data on the importance of nutrition for the future health in the susceptibility window of the first thousand days of life are numerous, a few studies have distinguished the pre-natal period from the post-weaning one. The data in the literature result mainly from animal studies and, therefore, further researches are needed in order to evaluate its effects on humans. In any case, these studies represent a starting point for understanding the important and complex metabolic impact of nutrition in such a critical period.

The aim of the study by Cho et al. [16] was to evaluate whether it was possible to modify pre-natal epigenetic alterations through dietary intervention starting from weaning up to 29 weeks, using an animal model. For the first time, they demonstrated the post-natal plasticity of the hypothalamus and its sensitivity to epigenetic influences during weaning. In fact, it emerged that nutrition in early stages of development would be able to modify the programming carried out in utero by the maternal diet. Therefore, the obesogenic phenotype and the reduced glucose response of the pups of rats, induced by the maternal diet, are modified through a high folate post-weaning diet. It was also highlighted that the observed effects were attributable to the change in the methylation status of the promoter of pro-opiomelanocortin (POMC), one of the main anorectic neurohormones.

The study by Zheng et al. [17] also studied the effect of the pre-natal and post-weaning diet in rat pups on the central nervous system. Specifically, the methylation status of DNA was analyzed in the hypothalamus, at the level of the POMC and its MC4R receptor (melanocortin receptor 4): fundamental components in regulating food intake and body weight. It was observed that the expression of POMC and MC4R at the hypothalamic level were both increased in puppies exposed, pre-natal and post-weaning, to diets high in lipids and sucrose (high fat and high sucrose diets), albeit with distinct mechanisms. Indeed, a hypo-methylation of the promoter was detected in the POMC, while the same effects were not observed at the receptor level; therefore, the cause of the increased receptor expression is to be attributed to other regulatory mechanisms. In any case, the metabolic effects in the offspring of this dual epigenetic marking of dietary origin were multiple: predisposition to obesity, glucose intolerance and insulin resistance later in life.

Moody et al. [18,19] analyzed the metabolic impact of high fat diets in early stages of development. The first study in this regard [18] analyzed the effects on the transcription regulation of the enzyme carnitine acyl-transferase-1 (Cpt1a) on rats. The results showed that there is an increase in the transcription of Cpt1a in the liver only when the exposure to diets with a high lipid content is prolonged over time, (lifelong diets) as the only pre-natal or post-weaning dietary contribution was not sufficient. It was also found that this transcriptional increase occurs through a very complex and highly coordinated mechanism. However, given the high complexity of the mechanisms involved, the authors themselves suggested the need for further investigations to compare the different epigenetic contribution of the pre-natal and post-weaning diet. In their second study, Moody et al. [19] compared the metabolic impact of high fat diets in the pre-natal (maternal diet) and post-natal (post weaning) period on mouse models. They showed that an early exposure to diets with a high lipid content is crucial in the DNA methylation profile in the liver. This results in unique methylation profiles at the level of insulin and phosphatidyl-inositol signaling pathways. These observations suggest that, probably, the impact of the pregnant woman’s diet induces de novo epigenetic changes in the fetus as a result of maternal metabolic alterations which, however, persist later in life. Nevertheless, at the postnatal level, there is a more direct mechanism of action, through which the diet directly alters the “conservation” of hepatic methylation of the mature organ (diets alter methylation maintenance in the mature liver). Thus, the timing of exposure to the diet determines distinct effects on the epigenome, within a specific genomic context.

The effects of the changes induced by the dietary micronutrients content were analyzed by Hoile et al. [20]. They performed a study on rats which showed that the metabolic effects of a high folate diet would seem to depend not only on exposure timing, but also on sex. The observed metabolic impact was related to the methylation status of specific CpG islands on which the transcription of key enzymes of glucose metabolism such as phosphoenolpyruvate-carboxy-kinase depends. Furthermore, these results suggest that the extent of these effects continues into adulthood. 

Even Bermingham et al. [21] also analyzed the epigenetic impact of micronutrients starting from weaning for eight weeks. From this study, using mouse models, it emerged that the dietary supplementation of folate and selenium in the diet of female puppies, whose mothers had been fed diets with a high lipid content and low in folate and selenium, can alter the hepatic methylation profile. This could represent the mechanism by which a dietary correction in the early age mitigates the effects of poor nutrition in the earlier stages, as during pregnancy and breastfeeding. However, more studies are needed in order to understand the role of epigenetic changes in mediating the observed changes and the health implications. 

A further contribution regarding the possibility of altering uterine epigenetic programming through dietary modulation of weaning is provided by the study by Sánchez-Hernánde et al. [22]. This analysis resulted from data on the different post-weaning diet-related weight gains observed in male rat pups whose mothers were subjected to a high multivitamin gestational diet. Thus, they demonstrated that a diet high in vitamin A starting from weaning can affect the plasticity of neuronal circuits in the postnatal period, reducing both food intake and weight gain in the post-weaning period. A modification of gene expression was detected both at the level of molecular pathways related to food intake, including POMC, and in brain reward pathways. 

The study by McKay et al. [23] also analyzed the correlation between post-weaning and perinatal diets on mouse models. They showed that a low-folate maternal diet and a high-fat diet starting from weaning lead to important changes in gene expression in the liver, surprisingly involving the same 642 genes. Furthermore, the effects were much more pronounced when the dietary insult came from both the maternal and the offspring diets. Although further investigations are needed in this regard, these studies have brought interest on the potential effects of dietary quality during weaning for the prevention of obesity and other metabolic diseases. Indeed, they suggest that even in humans, maternal dietary errors can be exacerbated by the incorrect feeding of the child; thus, predisposing them to a greater risk.

A further contribution on the influence of nutrition in critical growth periods comes from the study by Yoshie et al. [24]. They performed a study on murine specimens aimed at analyzing the late skeletal muscle repercussions of a high fat diet in early developmental stages (first to third month of life). They showed that epigenetic changes caused by the early nutrition mediate the alterations in skeletal muscle gene expression that occur in adulthood. This determines metabolic effects for the whole organism. These alterations are due to mono-methylations and pan-acetylations at the histone level.

These observations have led to the supposition that nutrition in early stages of development is one of the causes of the different individual responses to aging and that this may also be due to the alterations observed in skeletal muscle.

## 3. Flavor Experiences and Eating Behaviors

Some of the main obesity-related risk factors in children are represented by a mother’s pre-pregnancy excessive weight, an excessive maternal weight gain and high fasting plasma glucose during pregnancy. Other factors are also described in the literature, such as paternal overweight/obesity, a short period of breastfeeding and the early introduction of solid foods, although for the latter, to date, there is still no solid evidence [25]. Fogel et al. [5] analyzed these six aspects and for the first time related them to both eating behaviors and excessive weight in children at six years of age. The results highlighted how crucial eating behavior is in modulating the association between early risk factors and adiposity later in life. Indeed, in the study population, the strongest correlation between the presence of risk factors during the first thousand days and excess weight (adiposity outcomes) was found in children who ate larger portions of food, who ate faster and who consumed an elevated caloric intake. The study also highlighted the usefulness of precisely using these factors to select the children at a greatest risk to start, together with their families, a nutritional education. Indeed, the problems of parental excessive weight may be a reflection of poor eating habits in the family, responsible for the increased probability that children will experience excessive weight gain. In some studies, the effectiveness of correcting unhealthy eating habits has already been highlighted in the short term. 

In a pilot study conducted by Boutelle et al. [26], the preliminary positive effects in reducing non-hunger-related food intake were highlighted thanks to an attention modification training session in children. 

Faith et al. [27], on the other hand, documented the immediate effectiveness of a nutritional education program in families aimed at reducing the speed of food intake, an overweight-related risk factor. All this evidence underlines the importance of the family environment and food education, highlighting several critical issues. In fact, if, on the one hand, the eating style of the parents is crucial in determining the type and quality of the nutritional offer, the approach to nutrition is also important as it reflects the way in which the meal is lived. 

As highlighted by Johnson et al. [28], parental feeding styles and practice as well as parental beliefs are crucial in modulating children’s food choices.

In fact, the “Global strategy for infant and young child feeding” [29] of the World Health Organization (WHO) has already been promoting the responsive feeding since 2003 on the basis of the study by Pelto et al. [30]. Indeed, this research group proposed a collection of “complementary feeding behaviors best practices” obtained by combining the principles of psychosocial assistance with today’s knowledge in the nutritional field. The latest guidelines of the European Society for Pediatric Gastroenterology Hepatology and Nutrition (ESPGHAN) also promote the same type of approach to complementary nutrition [31].It, therefore, emerges that a responsive parenting approach to children’s nutrition, the absence of coercive methods and pressure, a constant good example and the commitment to expose children to a wide range of flavors, even the most particular ones without fear, are fundamental factors for establishing a correct relationship with food. 

In addition, there is the important gustatory path to forge the food preferences of children [32,33]. Now known is the existence of a very early gustatory perception that begins already in the uterine environment starting from the last trimester of gestation and continues during breastfeeding [34,35]. A recent systematic review of the literature by Spahn et al. [36] has in fact emerged that the evidence regarding the influence of the maternal diet on the food choices of children, during pregnancy and breastfeeding, is, respectively, limited, but consistent and moderate. The authors, therefore, stated that, to date, it is not yet possible to draw firm conclusions. In any case, this early exposure can contribute in modulating future food preferences; however, the period of complementary feeding plays a primary role. To date, the knowledge about children’s taste perception has considerably expanded and this is essential to provide adequate advices to parents and health personnel [37]. In this context, there is an important systematic review of the literature by Paroche et al. [37], which examined the role played by known developmental learning processes in the establishment of early eating behavior and food preferences. From this research, which analyzed several papers concerning children from the time of weaning up to 36 months, the presence of four developmental learning processes emerged. The first concerns the familiarization with flavor, texture and appearance of a food through repeated exposures that seem to be all the more effective the earlier they are performed and whose effects seem to last even in the long term. This method also seems to be effective in encouraging the future propensity of children to accept unknown flavors. The second concerns the imitation and the example of the family which, however, does not consider children as passive imitators but attentive selectors of the models to follow. In this context, the involvement of the child in the family meal is important and the long-term effects are remarkable in the future consumption of fruit and vegetables. There is certainly a strong influence between the availability of food and, therefore, the offers provided to children and the imitation itself; however, unlike familiarization, imitation affects the less precocious weaning periods. Associative learning has also emerged, which is a set of techniques that are based on the association between a less appreciated or rejected food with a known and appreciated food or with a highly energetic food. However, despite some shortcomings in the literature in this area related above all to other forms of associative learning, the data analyzed so far seem to agree that this type of conditioning offers no advantage over repeated exposure which is, therefore, to be preferred. Finally, there is the categorization of foods which, however, is less studied in this age group. It is also difficult to especially apply in early periods where there is a notable difficulty in distinguishing between what is edible and what is not. That stated, the literature seems to agree that children are likely to eat what is familiar to them.

As highlighted by Forestell et al. [38], children live within their own sensory world. They have specific preferences that tend to change both during childhood and during the transition to adulthood [38,39]. However, the response of the little ones to classic tastes such as sweet and bitter tends to be the same even in different cultures and this suggests that this behavior is innate in every human being. Specifically, the infant’s preferences for sweet taste have been known for years [39,40,41,42] and the intake has also been correlated to the activation of specific brain areas associated with positive stimulations [43]. In addition, the finding of the perception of sweet taste even in premature babies [44] supports the fact that already, during the intrauterine life, the fetus knows and appreciates this taste. In fact, this biological predisposition of all individuals from birth is certainly related to specific survival needs that have led humans to necessarily appreciate the sweet taste of breast milk from the very first moments of life to encourage breastfeeding. In addition, there is the need to adapt to the environment, in fact for our ancestors, the sense of taste represented the means to select nutrients and, therefore, the need for survival. This has probably favored a taste preference towards a sweet flavor in order to ensure the supply of sugary, energy-rich foods [42,43]. From the study by Nicklaus et al. [39], it emerges that sour and umami flavors are also generally well accepted during weaning. On the contrary, the bitter taste is not appreciated by the little ones; however, this contempt seems to arise with the passage of time [35,41,42].

In fact, Kajiura et al. [44] demonstrated that starting from two weeks of life, the rejection of this flavor becomes marked compared to indifference in the very first days of life. Even in this case, the possible reasons are to be placed within an evolutionary context according to which, at the time of our ancestors, bitter flavors could be associated with poisonous substances or poorly preserved foods [32]. In any case, recent studies have shown that the perception of bitter taste is more complex than previously thought. In fact, although a polymorphism of the *TARSR38* gene has been discovered, which determines a greater sensitivity of some individuals towards certain bitter tastes, this variation is related to only one of the twenty-five genes responsible for a bitter perception. Therefore, it does not appear sufficient to determine food choices also by virtue of the remarkable plasticity of the sense of taste in the human species. Consequently, if, on the one hand, an early knowledge (detection) of the sweet taste in the prenatal period allows the newborn to accept milk from the first moments of life, the opposite is also possible. That is, an early exposure to less traditionally accepted flavors is possible thanks to proper stimulation in the prenatal period, during breastfeeding and then with weaning [32,35,41]. In any case, as pointed out by Beauchamp et al. [34], early and repeated exposure to natural and healthy flavors is essential for promoting correct food preferences that will be maintained throughout life. The importance of early exposure also clearly emerges from the study by Forestell et al. [45] which underlines that even if during breastfeeding a greater tendency to the initial acceptance of some flavors is favored (provided that the mother has experienced them), during weaning, instead, there is a more total acceptance, if supported by repeated exposures. Hetherington et al. [46] focused on the importance of early and repeated exposures with a step-by-step approach in the introduction of vegetables during complementary feeding as well. These authors have highlighted how, in a context of low vegetable consumption in both adults and children, sometimes the implementation of a simple method (simple intervention) has important repercussions in the path of acceptance of some usually less accepted by children foods (vegetables). Nevertheless, sometimes the path of acceptance of a particular flavor passes through facial expressions of disgust and this can cause the parent to stop offering, which should instead encourage repeated offering, especially if the child continues to eat the food. In fact, in a recent study [47] on the subject, it was highlighted that although some aspects of the temperament of each child can influence the appreciation of some foods, parents should mainly evaluate the child’s willingness to continue with the tastings, allowing them to have different experiences with the same flavor in order to allow a complete acceptance of the flavor itself. In addition, the experience with all five senses represents a very important developmental stage for children that should not be denied [28].

Furthermore, as evidenced by Mennela et al. [48], children are very skillful at discriminating between flavors. The repeated exposure to a particular variety of foods, fruits for example, favors the acceptance of different types of fruits but not vegetables and the same occurs in the case of repeated exposure to vegetables which facilitates the consumption of other vegetables. The importance of exposure in children in promoting the appreciation of certain foods has been confirmed by numerous studies [39]; however, the number of necessary exposures is still controversial [32]. Indeed, while approximately 8–10 exposures were normally thought to be necessary, a recent study by Canton et al. [49] suggests that although repeated exposure remains the best technique to promote good dietary variability in children, the number of exposures required under the age of two of life could be inferior. In any case, neophobia is part of the development path of every child and strongly involves the sense of smell in an important way [39]. Nevertheless, in this context, the behavior of the family, the good example and the continuous proposal of a variety of foods [39] in the absence of pressure are decisive [28]. The most relevant key factors in flavor perception and food preferences in children are summarized in Table 1.

The data analyzed so far have shown that complementary feeding should be the starting point for a healthy dietary education, especially during the first thousand days of a child’s life and the mothers’ nutrition and mostly the weaning diet (Figure 2) represent a window of vulnerability that should be transformed into an opportunity [33,41].

## 4. Nutrimetabolomics 

Due to the plasticity of the early stages of development, incorrect food choices in this particular period can lead to a metabolic imbalance such as to significantly increase the risk of developing both metabolic diseases such as obesity, diabetes, atherosclerosis and hypertension and allergic manifestations [7] later in life. However, in humans, there are many variables, especially among the youngest [7], and, therefore, it is useful to elaborate specific dietary indications according to the individual nutritional phenotype [50]. Indeed, this represents the quantitative expression of the effects of the diet on the state of health or disease, depending on the particular genetic make-up [51]. It has been widely discussed that weaning is a crucial stage for the children’s health, on which future food choices also depend; however, nutrimetabolomics data regarding this particular period are very scarce. Safi-Stibler et al. [52] suggests a delicate metabolic role for complementary feeding. Indeed, from the comparative metabolomic analysis of a mouse model, greater metabolic repercussions in the offspring emerged following a high fat post-weaning diet (post-natal effect) compared to the effects caused by maternal obesity (pre-natal effect). Specifically, the metabolic status in adult male offspring was analyzed at the level of three tissues: the liver, hypothalamus and olfactory bulb. It was found that the high fat post-weaning diet affected the same three metabolites, 1,5-anhydroglucitol, saccharopine and ß-hydroxybutyrate, in all three tissues analyzed. Hepatic-produced 1,5-anhydroglucitol is related to glucose metabolism and diabetes [53] and its decrease in the model was associated with the diabetic phenotype found in adult male offspring. The decrease in saccharopine concentration may be the result of an increased amino acid catabolism that affects the production of acetyl-CoA and the tricarboxylic acid cycle or the production of ketone bodies such as ß-hydroxybutyrate, which are involved in energy production during fasting. In addition, the increase in ß-hydroxybutyrate can cause an alteration of neuronal activity, influencing behavioral outcomes and, therefore, also food intake. In any case, the major metabolic alterations were found in the liver and, to a lesser extent, in the hypothalamus and olfactory bulb. In fact, according to the KEGG (Kyoto Encyclopedia of Genes and Genomes) database, the post-weaning diet interfered, in the liver, with the metabolism of glutamine, glutamate, methane, alanine, taurine and hypotaurine and with the biosynthesis of ubiquinone and other similar terpenoids-quinones. Alterations in the metabolism of other amino acids, lipids, carbohydrates and vitamin B were also found, albeit to a lesser extent. The same database also reported, at the hypothalamic level, significant changes in both the metabolism of arginine, proline and aspartate, and the urea cycle. However, minor alterations were also found in the degradation of lysine, the metabolism of histidine and the biosynthesis of pantothenate and Coenzyme A. As regards the olfactory system, a significant alteration was observed in the metabolism of arginine and proline, in the degradation of lysine and in the biosynthesis of pantothenate and coenzyme A. Instead, the interpretation of the data with the SMPDB (Small Molecule Pathway Database) database, highlighted a further alteration at the level of the olfactory system due to the metabolism of alpha linolenic acid and linoleic acid. This highlights a greater plasticity of the liver in response to environmental changes, which may reflect its centrality in maintaining homeostasis. Furthermore, the functionality of hepatic metabolism is also crucial for other key organs, such as the brain, influencing its energy supply and various cell signaling mechanisms.

Chorell et al. [54] analyzed the metabolic impact of a probiotic, *Lactobacillus paracasei ssp. Paracasei* F19 on healthy infants during weaning, from the fourth to the thirteenth month of life. Both a decrease in palmitoleic acid and an increase in putrescine were observed. Palmitoleic acid, a monounsaturated fatty acid that derives from endogenous lipolysis, has recently been linked to visceral obesity in obese children by Okada et al. [55], while putrescine is a polyamine involved in cell growth and maturation. In studies on breast milk, it appears to be involved in the correct development of the baby’s intestinal barrier [56]. Unfortunately, Chorell et al. [54] did not take into consideration the diet of the children in question. Although these data support the centrality of bacterial flora in the metabolic well-being of children, it would be useful to analyze the concomitant impact of the weaning style. Indeed, in this context, correct nutrition is fundamental in favoring a healthy microbiota and it would, therefore, be useful to investigate this factor as well to evaluate its longer-term implications. In fact, the data on the follow-up [57] of school-age patients showed the total absence of metabolic feedback in subjects undergoing integration. In this regard, in the study by Zhang et al. [58] performed on obese children, it was observed that the dietary intervention designed to modulate the intestinal bacterial flora has a positive impact both in the reduction in simple obesity and that with a genetic component. Moreover, thanks to the metabolomic analysis, it was possible to detect that some specific circulating metabolites are produced only in the presence of intestinal microflora [57]. In fact, it has recently been shown that weaning represents a critical moment also for the intestinal microbiota as it decreases the strong influence of breastfeeding which is partially replaced by a food-related stimulation. Moreover, in this particular period of development, the maturation of the adaptive immune response is completed, also thanks to the antigenic load of the food itself. A delicate role, therefore, emerges for nutrition in the modulation of a nascent microbiota whose metabolites have important repercussions for development, growth and the immune system [59]. 

A further possible application of nutrimetabolomics in weaning comes from the study by Borresen et al. [60], aimed at analyzing the nutritional impact of three preparations for complementary feeding containing common kidney beans, cowpeas and a mixture of corn and soy. In fact, this study identified potential dietary markers related to the intake of legumes. Specifically, there are pipecolic acid and oleanolic acid, for the common bean, quercetin and alpha and gamma-tocopherolic acid, for the cowpeas and arabinose and serotonin for the corn and soy blend. Their potential use as specific markers in blood, urine and feces has, therefore, been hypothesized in future studies, in order to evaluate the effects, in terms of health, associated with the use of this vegetable protein source.

The metabolic impact of early nutrition on children’s health has recently emerged from the study by Hovinen et al. [61]. They analyzed, in a children population from 1.42 to 7.07 years old, the impact on the metabolism and on the status of micronutrients of the different types of diets: vegan, vegetarian and omnivorous. The results obtained from the metabolomic and biomarkers analysis in the blood showed that children who started a vegan diet since weaning have metabolic profiles and a micronutrient status different from the others, supporting the fact that even small intakes of animal proteins are sufficient to determine detectable changes in the metabolic pathways of children. Vitamin A deficiency states and borderline values for vitamin D have been mainly observed in vegan children. The values of total cholesterol, HDL and LDL, docosahexaenoic acid and essential amino acids were also significantly lower. Differences also emerged in the biosynthesis of primary bile acids and in the lipidomic profile (phospholipid balance). 

Furthermore, in the context of hereditary metabolic pathologies, the study by Burrage et al. [62] suggested that the metabolomic investigation is effective not only in the diagnosis of inborn errors of metabolism, but also in the monitoring of efficacy in clinical management and, therefore, also in dietary intervention, with the possibility of careful monitoring especially in critical periods of development.

The results of the metabolomics studies in weaning are summarized in Table 2.

## 5. Conclusions

It is now certain that nutrition in early development affects the risk of noncommunicable diseases later in life [1,63]. The high plasticity of the first thousand days of life makes this time window very critical [7]. In fact, this epigenetic programming of gene expression in various metabolic pathways may also largely depend on the maternal diet and post-natal nutrition. In many respects, the metabolic effects caused by the maternal diet during gestation are less controllable than those after birth [1]. In addition, the analysis of preliminary studies on animal models suggests that the postnatal diet may represent a window of opportunity to remedy deleterious prenatal epigenetic effects. In this context, adequate nutrition starting from weaning can become a valid remedy for reversing epigenetic programming. However, there are numerous data in the literature about the importance of adequate nutrition throughout life. The taste education for children during weaning and new methods of approaching nutrition by parents are, therefore, crucial to lay a solid foundation for the future. The new diagnostic investigation technologies, such as metabolomics, allow to concretize what has been analyzed in clinical practice. The recognition of early disease markers as well as food-specific metabolites will be the guide for an individualized and precise diet [64]. Metabolomics has already shown how it is possible to differentiate newborns both in terms of weight at birth and nutritional regimen [65]. This allows for the identification of particular groups of subjects at risk and the careful monitoring of adherence to dietary therapy can represent the basis for the medicine of the future: preventive and predictive.

## Figures and Tables

**Figure 1 nutrients-13-03351-f001:**
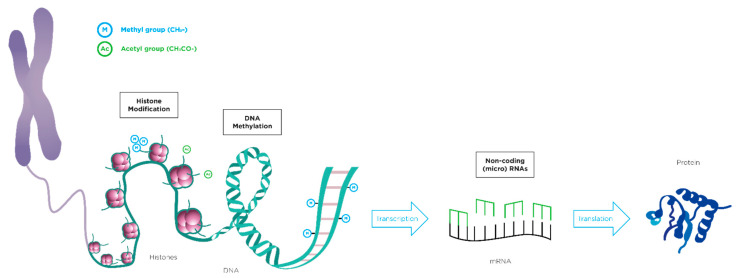
Epigenetics modifications: (1) classical way with an upstream control of gene transcription through methylation; (2) alternative post-transcriptional pathway which involves microRNAs.

**Figure 2 nutrients-13-03351-f002:**
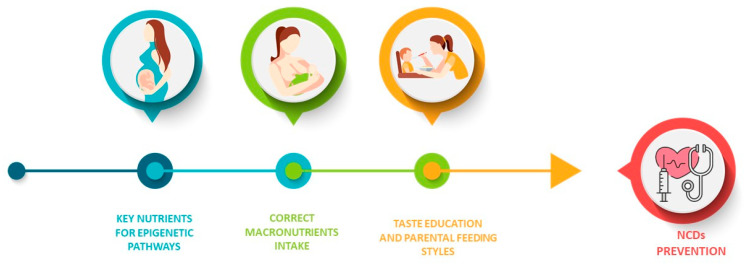
The crucial role of nutrition in the first 1000 days.

**Table 1 nutrients-13-03351-t001:** Key factors in flavor perception and food preferences in children.

Period of Interest	Food Exposure	Influence on Children Dietary Intake
Pregnancy	Amniotic fluid: some foods’ flavor can transfer to amniotic fluid and this fetus exposition can increase acceptance of these food during infancy/childhood (limited but consistence evidence)	The association between maternal diet during pregnancy and children’s dietary intake is yet to be proven due to the impossibility to isolate the influence of the maternal diet itself
Lactation	Breast milk: some food flavors can be present in breast milk in a time-dependent manner and infant can sense these flavors (moderate evidence)	No studies have analyzed the correlation between maternal diet during lactation and children’s dietary intake in the long term so far
Weaning experience	Repeated exposure to a variety of foods	Effective technique to learn about foods and to increase the acceptance of specific tastes that appear to have long-term effects on consumption (mostly from the beginning of weaning)
	Observational learning/family example	Effective technique with an impact on the eating behavior of children with long-term effects on consumption primarily of fruit and vegetable and to reduce food fussiness (mostly from 14 months)
	Associative learning	This type of conditioning seems to offer no advantage over repeated exposure which is, therefore, to be preferred
	Categorization	Difficult to apply especially in early periods of childhood: further studies are needed

**Table 2 nutrients-13-03351-t002:** Metabolomics studies in weaning.

Authors, Year	Patients	Sample	Characteristic	Technique	Main Results	Clinical Significance
Chorell et al. [54], 2012	20 infants, daily intake of cereals with or without the addition of *Lactobacillus paracasei ssp. paracasei* F19 (LF19)	Plasma	Human study (double-blind, placebo-controlled, randomized intervention trial)	GC-TOF/MS	↑ polyamine putrescine ↓ palmitoleic acid (c16:1) in the LF19 group	Feeding LF19 during weaning affects the metabolic profile
Hovine et al. [61], 2021	40 Finnish children vegan, vegetarian or omnivores	Blood samples	Human study (cross-sectional)	(TOF)-MS	↓ vitamin A, vitamin D, essential amino acids, docosahexaenoic acid and cholesterol (total, LDL and HDL); differences in primary bile acid biosynthesis and phospholipid balance in vegan children	Vegan children since weaning show remarkable metabolic differences compared to omnivores
Safi-Stibler et al. [52], 2020	Adult male offspring	Ground liver, intact hypothalamus and whole olfactory bulb tissues	Mouse model: (50% CD: 20% proteins, 70% carbohydrates, 10% fat vs. 50% HFD: 20% proteins, 20% carbohydrates and 60% fat)	LC–HRMS (non-targeted)	Three metabolites were affected by the post-weaning diet in all three tissues: ↑β-hydroxybutyrate ↓1,5-anhydroglucitol ↓saccharopine	Post-weaning diet had a significant impact on the abundance of metabolites in the liver and, to a lesser extent, in the hypothalamus and whole olfactory bulb

Abbreviations: ↑ (increase), ↓ (decrease), LF19 (*Lactobacillus paracasei ssp. Paracasei* F19), GC (Gas Chromatography), MS (Mass Spectrometry), TOF (Time of Flight), LC (Liquid Chromatography), HRMS (High Resolution Mass Spectrometry), LDL (Low Density Lipoproteins), HDL (High Density Lipoprotein), CD (control diet), HFD (high-fat diet).

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
