# Peer review of "Epigenetics and Modulations of Early Flavor Experiences: Can Metabolomics Contribute to Prevention during Weaning?"

_nutrients, 2021, doi:10.3390/nu13103351_

Round 1

Reviewer 1 Report

The present review discusses changes in the metabolomics and flavor preferences in different weaning practices. It is well written and backed up.

One question remains concerning the study selection by the authors. There seem to be many more studies conducted on the subject that were not included in the review.

Table 1, in the study by Safi-Stibler et al., what was the diet provided to the rats?

Table 1: were the human studies RCTs? What was the design?

The figures require some improvement in terms of quality.

One table with studies on changes in flavor preferences would also be an asset.

Author Response

Reviewer 1

The present review discusses changes in the metabolomics and flavor preferences in different weaning practices. It is well written and backed up.

One question remains concerning the study selection by the authors.

There seem to be many more studies conducted on the subject that were not included in the review.

      Answer. Thanks for the suggestion, the weaning period is a very delicate moment so there are not many studies about this topic. Thus, for what concerns the studies on epigenetics we did our best in reviewing the literature. However, we proceeded to cite one of the first pioneering studies performed by Waterland et al [15]. As regards metabolomics, studies concerning this topic are very scarce, while for the development of taste and methods of approaching complementary nutrition we have cited and analyzed 11 new studies about the subject[29-31,33,35-37,39,41,42,46].  

Table 1, in the study by Safi-Stibler et al., what was the diet provided to the rats?

Answer. Thanks for the comment, we have illustrated the details of the diet provided to the rats in the table.

Table 1: were the human studies RCTs? What was the design?

Answer. Thanks for the comment, we have further described the design of the study in the table.

The figures require some improvement in terms of quality.

Answer. Thanks for the comment, we have improved the quality of the figures according to the reviewer’ suggestion.

One table with studies on changes in flavor preferences would also be an asset.

Answer. Thanks for the suggestion, we have added a table about key factors in flavor perception and food preferences in children.

Reviewer 2 Report

The authors submitted a comprehensive review of the window of cell plasticity (first 1000 days of life), epigenetic modifications, the influence of nutrition, and taste development. They focus on the weaning period and speculate that the use of metabolomic profiling will allow the planning of an individualized and precise diet. However, there are no studies in the literature with metabolomic profiling as the primary outcome to study different weaning diets of infants. It might be helpful to focus more on the present research gaps and future studies needed to prove that metabolomic profiling can really contribute to select a healthy personalized weaning diet for an infant/toddler.

some minor comments:

  • the authors might consider modifying the title of their review because only 25% of the text is on metabolomic profiling
  • epigenetics:  the groundbreaking work of Waterland RA at the startup of epigenetics (folic acid supplementation during pregnancy in genetically obese animals) should be mentioned
  • metabolomic profiling: the Finnish study on vegetarian/vegan/mixed diet (49) was done in children aged 3.5 years. This is not the weaning period. The study of Chorell (42) evaluates a nutritional supplement (Lactobacillus casei) which is not part of an average weaning diet (EFSA). It is well established in many industry-sponsored studies that supplementation with different probiotics, prebiotics, and HMOs modifies the microbiome and thus the metabolomic profile. However, metabolomic profiling was never the primary outcome variable and then used as the basis for recommendations during the weaning period 
  • The study on legumes and metabolic profiling (48) was done in an African rural population where malnutrition (stunting, wasting) usually starts during the weaning period (inadequate weaning foods). Results might not be representative of infant populations during the weaning period in developed or emerging countries.
  • conclusions: the authors mention their own studies (52,53) as arguments that metabolic profiling can be an important tool during the weaning period. A study in neonates (preterm- vs term) during the first weeks refects early differences in nutrition (parenteral/enteral) and health but is not related to weaning food 

Author Response

 Reviewer 2

1. The authors submitted a comprehensive review of the window of cell plasticity (first 1000 days of life), epigenetic modifications, the influence of nutrition, and taste development. They focus on the weaning period and speculate that the use of metabolomic profiling will allow the planning of an individualized and precise diet. However, there are no studies in the literature with metabolomic profiling as the primary outcome to study different weaning diets of infants. It might be helpful to focus more on the present research gaps and future studies needed to prove that metabolomic profiling can really contribute to select a healthy personalized weaning diet for an infant/toddler.

some minor comments:

the authors might consider modifying the title of their review because 
only 25% of the text is on metabolomic profiling

Answer. Thanks for the suggestion. Taking into consideration your remarks regarding the fact that “there are no studies in the literature with metabolomic profiling as the primary outcome to study different weaning diets of infant” we can modify the title accordingly:

EPIGENETIC AND MODULATION OF EARLY FLAVOR EXPERIENCES: CAN METABOLOMIC CONTRIBUTE TO PREVENTION DURING WEANING?

2. epigenetics:  the groundbreaking work of Waterland RA at the startup of epigenetics (folic acid supplementation during pregnancy in genetically obese animals) should be mentioned

Answer. Thanks for the suggestion. We added the paper in the reference list.

  • metabolomic profiling: the Finnish study on vegetarian/vegan/mixed diet (49) was done in children aged 3.5 years. This is not the weaning period.

Answer. Thanks for your observation. Although the study by Hovinen et al. was performed on vegan children with an average age of about 3.32 years, the authors specify several times that the sample under analysis follows that particular dietary restriction starting from weaning. This data is reported not only in the description of the study population where it is also indicated that the age of vegan children is between 1.75 years and 6.34 years but in the discussion section as well. We thus considered that this clarification was important and therefore that the observed alterations are also the result of a totally vegetable based weaning. However, probably by writing "since weaning" we were not sufficiently clear and precise, so we have included in the text the clarification regarding the age of the children at the time of the analysis.

3. The study of Chorell (42) evaluates a nutritional supplement (Lactobacillus casei) which is not part of an average weaning diet (EFSA). It is well established in many industry-sponsored studies that supplementation with different probiotics, prebiotics, and HMOs modifies the microbiome and thus the metabolomic profile. However, metabolomic profiling was never the primary outcome variable and then used as the basis for recommendations during the weaning period.

Answer. Thanks for the comment. Our goal was by no means to consider the use of probiotics as part of weaning or even delve into their benefits. However, we thought it could be interesting to evaluate the effects of the microbiota on some metabolites (putrescine and palmitoleic acid) in the weaning period in the light of new knowledge regarding their impact on the metabolic health of children. In fact, we specified that the study that we mentioned did not take into consideration the diet followed by the participants, underlining how this could be a very interesting data. In support of this, we wanted to emphasize that the follow-up did not show long-term effects, further confirming that the dietary impact should also be taken into consideration because a healthy dietary routine throughout life certainly has important repercussions on the microbiota and therefore on the metabolic health of children, as pointed out by Zhang et al. Nevertheless, for the sake of clarity, we have further reiterated these observations in the text.

4. The study on legumes and metabolic profiling (48) was done in an African rural population where malnutrition (stunting, wasting) usually starts during the weaning period (inadequate weaning foods). Results might not be representative of infant populations during the weaning period in developed or emerging countries.

Answer. Thanks for your observation. We did not consider the limits of the population of this study to be important, as the data we considered only concern the identification of possible dietary biomarkers of the consumption of legumes. We have in fact underlined how the possibility of using dietary biomarkers associated with the consumption of vegetable proteins during the weaning period is potentially useful for evaluating, in future studies, their impact on health in the short and long term.

5. conclusions: the authors mention their own studies (52,53) as arguments that metabolic profiling can be an important tool during the weaning period. A study in neonates (preterm- vs term) during the first weeks refects early differences in nutrition (parenteral/enteral) and health but is not related to weaning food

Answer. As experts in clinical metabolomics applied in neonatology, given the number of experimental studies published in recent years by our group, we are well aware of the current limitations of this technology. But at the same time, we are also convinced of its potential, which, in the future, will allow us to evaluate in a concrete way how preferable a type of weaning is for each child. The study mentioned by the reviewer was included in our paper for the following reasons. On the one hand for completeness of information regarding the current application of metabolomics in the field of infant nutrition, on the other hand it is an example of how this ‘omits’ science can be useful in monitoring metabolism of the child in critical periods of development such as weaning.

Round 2

Reviewer 2 Report

Minor comment revised title Instead ofEpigenetic  Epigenetics? instead of Metabolomic - metabolomics or metabolomic profiling.